# How and Why Diets Change Post-Migration: A Qualitative Exploration of Dietary Acculturation among Recent Chinese Immigrants in Australia

**DOI:** 10.3390/nu14173573

**Published:** 2022-08-30

**Authors:** Sarah D. Lee, Nicole J. Kellow, Catherine E. Huggins, Tammie S. T. Choi

**Affiliations:** Department of Nutrition, Dietetics and Food, Faculty of Medicine Nursing and Health Sciences, Monash University, Notting Hill, VIC 3168, Australia

**Keywords:** Chinese, dietary acculturation, dietary change, qualitative

## Abstract

Chinese immigrants living in Western countries are at increased risk for cardiometabolic diseases. Dietary acculturation has been implicated as a potential contributor, but little is known about why diets change post-migration. The purpose of this qualitative research study was to explore how and why diets change post-migration for Chinese immigrants living in Australia. Eleven participants undertook semi-structured interviews exploring and comparing their diets when they lived in China to their post-migration diets. Thematic analysis revealed that participants exhibited changed social structures of meal preparation, and made unacknowledged dietary changes, such as recipe modification, to maintain their traditional Chinese diet post-migration. Implications of both deliberate and unrecognized dietary changes post-migration include connections to increased risk for metabolic disease post-migration.

## 1. Introduction

Chinese immigrants living in Western countries have a four-fold higher prevalence of coronary heart disease (CHD) than those who remain in China, as well as a significantly increased risk of type 2 diabetes mellitus (T2DM) and other cardiovascular disease (CVD) risk factors [1,2,3]. As CVD and T2DM are among the leading causes of death worldwide, an examination of what contributes to this increased risk among Chinese immigrants is warranted. While the reasons for this increased risk are not entirely understood, dietary acculturation is hypothesised/proposed as a contributing factor [4,5]. Dietary acculturation is the change in eating habits that occurs when a person moves away from their country of origin to a new host country and begins to adopt the new eating patterns of their host country [6]. Dietary acculturation in Western countries may include increased intakes of ultra-processed foods high in sugar, saturated fat, and sodium, which may contribute to an increased risk of chronic disease [7]. A review examining dietary acculturation of Asian Americans found that Chinese immigrants increased their consumption of Western foods and reduced their consumption of traditional Chinese foods after being in America for more than 10 years [8]. However, dietary acculturation is not a linear process. Rather, it is a complex, poorly understood phenomenon that involves both the retention of some dietary behaviours while also incorporating new foods and eating habits from the new country [6]. Based on research conducted in Chinese immigrant populations [6,9], Satia [10] developed a model to help explain the process of dietary acculturation, acknowledging that immigrants come with pre-existing socioeconomic, demographic, and cultural factors, and then through exposure to the new host culture, they experience changes in psychosocial factors and taste preferences, as well as changes in environmental factors, such as food availability, which together result in their new dietary intake pattern [10].

While epidemiological studies have identified risk factors for CVD [11,12,13,14,15], qualitative research is needed to unravel the complex, interacting system of factors that result in dietary change and subsequently increased disease risk for Chinese immigrants. It is inevitable that some dietary change will occur when Chinese people move to a new country, because the food environment and availability of ingredients is different. Still the fundamental reasons why diets change are poorly understood. A recent scoping review highlighted the lack of standardized assessment tools for dietary acculturation as a barrier to measuring how dietary acculturation occurs and how this phenomenon might be contributing to observed increases in chronic disease prevalence for migrants [16]. This review also suggested a need for further qualitative exploration of the experiences of dietary acculturation in order to better understand the factors that influence dietary decision-making processes of migrants [16]. The bulk of the literature exploring dietary acculturation in Chinese migrants has been conducted using cross-sectional surveys, most of which did not use validated tools to measure dietary acculturation and lacked depth of understanding the process of dietary change [16]. Understanding why diets change post-migration enables the development and implementation of appropriate intervention strategies to address increased diet-related disease risks and promote quality of life for migrants. A large longitudinal cohort study, conducted 20 years ago, examined the food intake behaviours of Chinese-born immigrants in Australia and found that, over a period of 8 years on average, Chinese Australians increased their consumption of protein and fat while decreasing their carbohydrate consumption [17]. However, this study did not explore the reasons behind these dietary changes nor examine the relationship between these dietary changes and the risk of cardiometabolic disease. As there is currently a gap in the literature exploring why diets change post-migration for Chinese immigrants, it is important to understand factors contributing to dietary acculturation in order to better understand how to promote dietary changes that reduce chronic disease risk factors. Therefore, the purpose of this study was to qualitatively explore Chinese immigrants’ experiences of dietary acculturation in order to better understand how and why their diets change post-migration.

## 2. Methods

### 2.1. Study Design

This exploratory qualitative descriptive interview study was conducted with adult Chinese immigrants to Australia to better understand the complex process of dietary change for Chinese immigrants. The study protocol was approved by a university human research ethics committee (project ID: 24622).

This research was guided by constructivism whereby individuals construct meaning of their experiences within their social context and values system, and multiple realities of the same event can exist [18]. The semi-structured interview questions were designed to draw out participants’ experience of dietary acculturation, and thoughts and values underpinning dietary change. The first author is a bilingual (Chinese and English) researcher who has intimate knowledge of Chinese culture and is also an immigrant so was able to relate to the experiences of participants and build rapport.

### 2.2. Participants

Purposive and snowball sampling methods were used, aiming for diversity of sample with variation in age, gender, and length of residence in Australia for adult Chinese immigrants. Eligible participants were Chinese immigrants over the age of 18, who were born in Mainland China, had migrated directly to Australia less than 10 years prior, and spoke either English or Mandarin. Those born in Hong Kong, Macau, South East Asia or Taiwan, as well as those who had been living in Australia for more than 10 years, were excluded. The cut-off point of 10 years was used, as the previous literature on Chinese migrants has found that diets tend to stabilize after 10 years post-migration, and that the health status of migrants tends to converge with that of the host population after 10 years [5,19,20]. We wanted to capture what happens in the early stages of dietary acculturation post-migration, as well as to ensure that participants would still be able to recall their previous dietary habits in China, which is why 10 years was chosen as the cut-off point. Participants were recruited through WeChat advertisements, a registry of interested individuals from a previous study of Chinese migrants, and through word-of-mouth snowballing from existing participants.

Participants expressed interest by completing an online screening questionnaire hosted on the REDCap (research electronic data capture) platform. The web-based REDCap is a secure software platform designed to support data capture for research studies [21,22]. The first author contacted eligible participants via phone call, WeChat, or email to explain the study aims and protocol. For those who agreed to participate, a Zoom meeting was organised. Verbal informed consent was obtained prior to the interview.

### 2.3. Setting

Chinese people represent the second largest group of non-English speaking migrants in Australia, with approximately 651,000 people born in China currently residing in Australia [23]. Globally, there are over 50 million people of Chinese descent living outside China, making it the second largest source country for migrants to the United States and Canada after the United Kingdom [24]. This study was conducted in a metropolitan area of Australia where Chinese immigrants make up approximately 6% of the total population and Asian food markets and restaurants are plentiful and easily accessible [25].

### 2.4. Data Collection

The dietary acculturation model proposed by Satia [10] was utilized to inform the line of inquiry of the interview questions. Data collection included semi-structured interviews, and an online demographics survey was hosted on the REDCap platform.

A semi-structured interview guide was developed and pilot-tested with two members of the target population before being further refined for clarity to answer the research question. The final interview guide and question logic can be found in Table 1. Though the original plan was to conduct face-to-face interviews, due to the COVID-19 pandemic, interviews were conducted via an online video communication platform (Zoom Video Communications Inc., platform San Jose, CA, USA) in participants’ preferred language (Mandarin Chinese or English). Interviews were conducted by two researchers in tandem, who took turns asking questions, both with bilingual language abilities in order to minimize language comprehension issues and researcher bias. The first author has a background in nutrition with experience conducting qualitative research. Both interviewers also had a good understanding of Chinese culture and food habits, which allowed for better understanding of participant’s cultural perspective and insider knowledge of the impact of migration on dietary habits. Interviews were video and audio recorded for verbatim transcription. Interviews that were conducted in Chinese were transcribed in Chinese prior to being translated into English by one of the interviewers. Interviews lasted between 30 min and 1.5 h, and recruitment of participants continued until data saturation had been reached, defined by the point at which new data did not add further to the story and it was no longer productive to continue recruitment [26].

Prior to the interview, each participant completed a survey to collect demographic data including, age and length of residence in Australia, as well as some brief anthropometric information. See Appendix A for the full questionnaire.

### 2.5. Data Analysis

Interviews were transcribed verbatim and then subsequently translated into English by one of the interviewers for analysis. A subset of interview translations was verified by a second researcher (T.C) with bilingual language abilities to ensure accuracy of translation. Thematic analysis was conducted using the approach originally developed by Braun and Clarke [27] and refined into definitive steps by Nowell et al. [28], whereby the researcher familiarized themselves with the whole dataset prior to creating a set of codes and then searching for themes among the codes. Coding was conducted manually by the primary researcher (S.L), and a subset of transcripts were double-coded by a second researcher with bilingual capabilities (T.C) to ensure congruence [29]. Themes were developed and refined through discussions with the research team. Themes were generated inductively from the interview transcripts [27]. Themes underwent several iterations and stages of definition as well as investigator triangulation, as defined by Patton [30], whereby two researchers coded and sorted codes into categories before discussing and agreeing on the breadth of themes and revising prior to final naming of themes. Demographic data from the survey was used to provide context to the interviews and a snapshot of the characteristics of participants participating in the research study.

## 3. Results

A total of 11 participants were interviewed (*n* = 3 males and *n* = 8 females), ranging in age from 22–68 years (median age = 43 years), with length of residence in Australia ranging from 1–8 years. All participants were living in urban areas with household sizes ranging from 1 to 4 persons. Participants were most commonly employed part-time and had obtained a level of education equivalent to a Bachelor’s degree or higher. All participants originated from coastal cities in China. See Table 2 for full demographic results. Four main themes relating to dietary change post-migration were identified, namely the changed social structures of mealtimes and meal preparation responsibilities, that recipe modification is unacknowledged dietary change, that embracing new foods is purposeful dietary change, and that resistance to dietary change occurs. Exemplary quotes were chosen that best represent each theme while providing the range of views of the participants.

### 3.1. Theme 1: Changed Social Structures of Mealtimes and Meal Preparation Responsibilities

As participants discussed their dietary behaviours post-migration, it appeared that many of their dietary habits reflected the two cultures (Chinese and Australian), indicating that some acculturation had occurred. Participants attempted to maintain their traditional Chinese eating patterns because this is what they had grown accustomed to and were familiar with. At the same time, they were faced with challenges in preserving these eating patterns in the food environment of Australia because of different lifestyles, family structures, and food availability. Therefore, sometimes the structure of meals throughout the day remained the same while the specific foods changed. This was evident through the way participants held onto their home country eating behaviours in their new environment. In China, food was usually purchased on the streets, provided within the workplace, or prepared by extended family within the home, and was very easily accessible, cheap, and of good quality. For example, one participant said “*in Australia I need to cook for myself. But in China we have more options for…meals…we can [pay] less and [have] more options…and it’s more convenient for me*”* (Participant 4, female, 27 years).* In Australia, on the other hand, participants mentioned Chinese food was available for purchase outside the home but was expensive, not easily accessible, less authentic, and did not taste the same as they were used to in China, leading them to search for alternative ways to maintain their cultural eating habits, such as preparing the foods themselves. “*In China, for the most part [my] parents, or my husband’s parents purchase and cook food. [In Australia] both [of us]-referring to husband and self—will cook, half and half*” *(Participant 7, female, 36 years).* Despite their adaptations and coping strategies, the lack of family support when it came to meal preparation affected participants’ choice of foods, often leading to more convenient ‘Western’ foods being chosen as these were easier to prepare.

In China, the workplace usually provided food (breakfast and lunch) for staff, but in Australia they now have to prepare all meals for themselves, which also impacts the type of foods they prepare. “*So in China, because I worked for the state government, so usually I have my brekky and my lunch from the canteen of the government department. So…they have a variety [of foods]…they are like Chinese style but…you can just choose whatever you want, and they are all free…what I had in China…also about the [food in] China…[over] there I had something more healthier, but here just…find a way to prepare it quicker…it’s not fresh because here usually I use the microwave to warm it but…in China you can get the fresh food*” *(Participant 8, male, 30 years)*. During their time in Australia, most participants reported that they personally cooked most of their meals at home (primarily midday and evening meals) compared to when they were living in China, where they frequently consumed two or three meals per day that were cooked by others or outside the home. The same was true for food shopping as many participants did not shop for food in China because they were not responsible for food preparation, while in Australia they were required to take on the roles of both cooking and food shopping. This sometimes resulted in reduced dietary variety. “*Over here…you have to make [food] yourself. It is not easy to [purchase foods]. It’s like whatever I make, [like when I] make fried radish patty, sticky rice in lotus leaf, sometimes [I] will make [these] when I am in the mood, but in Guangzhou I will not make it, because [I can purchase] next door. [And in Australia when] you make too much…then…it’s like [I have to] eat it today, eat it tomorrow, eat it the day after that, [and] you also don’t want to*” *(Participant 11, female, 59 years).*

In terms of changes to the types of foods eaten at specific mealtimes, the breakfast meal appeared to be the first meal to change post-migration, and the participants interviewed had almost entirely changed to Western-style breakfast foods (bread roll, toast, milk, and coffee). This is in contrast to dinner, which was primarily maintained in the Chinese style (rice, soup, and stir-fried meat/vegetable dishes). Participants reported that in China they previously ate a Chinese style breakfast (steamed buns, noodles, and congee), which they either purchased outside of the home on their way to work or which was provided to them by their workplace. The reasons provided for changing their breakfast to Western-style were that the Western breakfast foods were just as quick and convenient in Australia as the Chinese-style breakfast was in China. It was important for participants to have a quick and convenient breakfast because that is what they were used to in China. In order to maintain this well-established behaviour, participants were limited to the foods that were available and easily accessible in Australia, which happened to be Western-style foods. “*I take my brekky less healthy than what I had in China because…yeah…here it’s pretty quick, I just, like drive through [the fast food outlet] and get what you want…and then consume it…but in China, you can have a variety of choices because the work provides canteen…I don’t think there’s a lot of change about my dinner [except], I would say…what I mentioned before, the ingredients. I have to change it because I cannot find those ingredients here*” *(Participant 8, male, 30 years).*

Overall, participants discussed trying to maintain the structure of meals with a preference towards eating Chinese-style food when possible, but where others previously prepared the food, participants were now required to shop and prepare food themselves due to lack of family support and cooking skills. It seems the lack of family/social support in meal preparation had the most effect on the breakfast meal.

### 3.2. Theme 2: Recipe Modification Is Unacknowledged Dietary Change

Many participants expressed a preference towards eating their familiar, traditional Chinese foods, and undertook a number of strategies in order to maintain their cultural eating habits. One of the ways participants expressed this was by substituting or omitting ingredients in traditional Chinese recipes when they were not available in the host country. Participants felt that the lack of availability of certain ingredients in Australia was only a minor inconvenience and they were able to easily substitute most Chinese ingredients with a suitable alternative. For example, in terms of types of fish, river fish were swapped with ocean fish or fresh was swapped with frozen varieties. “*Originally, [in China I] possibly ate relatively more freshwater fish, although (right now) [I am] possibly all eating salmon [instead]*” *(Participant 7, female, 36 years)*. Certain seafood items that were specific to their hometown in China were omitted from the recipe where there was no substitute available. “*For example, some seafood…[because] in my hometown, it is close to the sea, and so there are some special seafoods, …I can’t buy here. And so I would cook very little seafood here, and have turned to protein sources from meat products*” *(Participant 1, male, 30 years).* This flexibility and recipe modification exhibited by participants led many to perceive that their diet had not changed significantly since migration. For example, one participant responded when asked how their diet had changed post-migration, “*Not much change. Similar. [I] will purchase whatever is available. It is not that…[I] am very picky about eating, [I am not like]: “[I] have to eat like this!*” *(Participant 11, female, 59 years).* The reasons why participants did not perceive recipe modification as a form of dietary change was because they felt it was a necessary part of moving to a new country. For example, one participant said, “*[you] have to eat according to what Australia [has to] eat, that is the choices…for example, there is only…these [varieties] of lettuce, then I will only [be able to] eat these [varieties]*” *(Participant 11, female, 59 years).* Another participant mentioned, “*if you decide to like work here or even live here, you have to get used to the culture here so you have to choose the food here (Australia) not like always be thinking about the food back in your country (China) so if you cannot get used to the food, then what about the rest of your life here, you have to go back home (China)*” *(Participant 8, male, 30 years).* These sentiments were echoed by the majority of participants interviewed, as well as feelings that the changes they had to make to their diets were not very significant overall and were just part of moving to a new country. Reasons for needing to modify recipes were generally related to lack of availability, freshness or cost of particular Chinese ingredients. “*There are some…veggies that’s more expensive in Australia. So we probably eat…other veggies instead of the expensive ones*” *(Participant 10, male, 68 years).*

Participants demonstrated flexibility towards dietary adaptation when they would substitute locally available ingredients into traditional Chinese recipes and did not feel like this made too much difference overall to their diet. This resulted in unacknowledged dietary changes post-migration, as participants perceived that they were still eating largely in the same manner as when they lived in China.

### 3.3. Theme 3: Embracing New Foods Is Purposeful Dietary Change

It was evident through the interviews that many of the participants also began to openly embrace dietary change by trying new foods and cooking Western-style foods themselves at home, including cakes, breads, steak, and salad. Those that migrated in their early 20s were more likely to be open to accepting of trying new foods compared to those that migrated at a later age and, consequently, spent a smaller percentage of their adult life in Australia. “*For me…[there is] no problem. I am very accustomed. So…it is all quite convenient. If I were to, if I were to solely eat Western foods, I would also be able to accept it, I am like this*” *(Participant 11, female, 59 years).*

Snacking was an opportunity to integrate more Western-style foods into the diet. Snacks are widely available in Australia, and participants trusted that the pre-packaged snacks had been prepared in a safe manner compared to China where there is a lot of distrust in the safety/transparency of the food industry. “*Overall it is still similar to [my] original [eating habits], however the consumption of snacks has increased…[I] always want to purchase snacks and try them…in China there may be worry towards the safety of foods. and then I will not go purchase them often. But over here, I feel that there are many choices, and then I am not too worried about food safety*” *(Participant 7, female, 36 years).* Participants also mentioned that they liked trying new snacks as they didn’t have to worry about how to cook them, and they felt it was a good way to experiment with Australian foods.

Participants also discussed cooking Western-style foods at home after migrating because they wanted to try to new recipes and foods that they saw around them. “*After I came to Australia, I was able to copy [the recipes] at home, and then cook Italian pasta, make bread, make cake…and then…all cook/make [myself]*” *(Participant 6, female, 43 years).* Another participant said, in regard to how their diet had changed post-migration, “*[after] coming here we increased [consumption] of Western foods…usually every week we will make Western food once. This is also a change from the past*” *(Participant 10, male, 68 years).*

Overall, most participants expressed willingness to try new foods in Australia and began to change their eating habits to include more Western-style foods both in the form of store-bought food as well as cooking them at home themselves. Reasons for making these dietary changes related to wanting to embrace their new home, as well as curiosity about new foods.

### 3.4. Theme 4: Resistance to Dietary Change

While most participants expressed willingness and curiosity towards trying new foods and even incorporated them into their daily diet, some participants described a resistance to changing some of their dietary behaviours post-migration for a variety of health and personal reasons. Reasons for resisting changes in diet were usually related to personal health beliefs and habits that had been ingrained for a long time and, therefore, were not easily swayed. “*Because I feel that I have always eaten this way, and have not felt discomfort, or I haven’t felt that I am undernourished. It is like this*” *(Participant 2, female, 65 years).* Participants who migrated at an older age were more likely to maintain their well-established eating habits and had the time and resources to do so. “*Our eating habits [referring to her daughter], she and I together, because she is young, I anyhow, am of greater age now…her staple foods are kind of opposite to mine. Because, I have lived there (China) for a long time…[my eating] habits have mostly been developed there. And when I moved here, usually for the most part [my eating habits] are [unchanged]*” *(Participant 2, female, 65 years).* Another participant expressed that they would never consider changing their habit of drinking hot water instead of cold “*Another point, all this time what has not changed for me is, drinking hot water, drinking warm water…I think iced water, especially in the summer, if you are considered [someone who] exercises a lot and drinks iced water, [it is] especially damaging*” *(Participant 3, female, 54 years).* Another participant offered one reason why some immigrants resist changing their diets post-migration, saying that, “*Although people are immigrants, their stomachs are still Chinese. And so, to change these eating habits is actually quite difficult*” *(Participant 1, male, 30 years).*

Other participants reported the availability of Chinese grocers in Australia allowed them to continue eating their preferred traditional foods. Additionally, some participants maintained their home country eating patterns because they were more familiar with the Chinese cooking methods/ingredients, whereas they did not know how to cook many of the vegetables in Australia. Some preferred to limit trying new foods to when they were eating out at restaurants, as they did not have to worry about preparation of the food. “*[I] do not intentionally try [new foods]. Because some Australia…some…those vegetables, green vegetables…some I cannot even say the name, I also don’t know how to cook with it. Some foods…but…[I will eat those foods] out [in restaurants]…But those [foods] I do not know how to cook. So I rarely purchase them*” *(Participant 11, female, 59 years).*

Overall, there were varied reasons why certain aspects of the diet did not change post-migration, including personal taste preferences, familiarity with cooking methods for Chinese foods, readily available Chinese ingredients, and ingrained habits and traditional health beliefs. If participants held strong beliefs about the health benefits of the way they were eating in China, then they did not see why they should change these eating habits post-migration even if it caused some inconvenience for them.

## 4. Discussion

The aim of this study was to explore the experience of dietary acculturation for people of Chinese-origin living in Australia who had migrated less than ten years prior. Participants described dietary behaviours post-migration that reflected changes primarily in the social structure of meals and food preparation responsibilities, as well as unacknowledged dietary changes through recipe modification. Dietary changes occurred due to lack of family support and the reduced accessibility of traditional Chinese food ingredients, as well as curiosity regarding new foods available their new home country. Overall the findings of this qualitative study have expanded on the model of dietary acculturation proposed by Satia [6], suggesting that dietary acculturation is impacted by multiple levels of influence including personal (taste preference and beliefs), interpersonal (family and peer support) and environmental (availability of foods, lifestyle changes) factors which are different to the factors impacting dietary choices in their home country. Through the interviews, it became clear that some dietary acculturation occurs unconsciously, as certain coping strategies are not recognized as dietary change. In navigating a new food environment, Chinese migrants actively seek ways to maintain their traditional eating patterns while also remaining open and willing to incorporate new foods, reflecting for the most part a bicultural eating pattern [6]. What the current study adds is that Chinese migrants adopt a bicultural eating pattern due to convenience, time for food shopping and preparation, as well as personal health beliefs and food literacy. Unacknowledged dietary changes occur through omitting and substituting ingredients in Chinese recipes while dietary change also occurs deliberately through including Western-style meals, particularly breakfast, and snacks, in their diet. Another key dietary change observed in this population was a shift in food procurement and meal preparation responsibilities due to changed family structures and lifestyles post-migration.

Often, for Chinese migrants, their diets appear caught between the East and the West in a sense that they try to maintain their traditional eating patterns in their new host country, but find it difficult due to changes in lifestyle and the food environment. These findings are also consistent with a study on dietary patterns of Chinese immigrants living in Spain, which found that Chinese immigrants both maintain their Chinese eating habits while simultaneously incorporating host-country eating habits and foods into their diets [31]. There was general consensus that most Chinese food ingredients are readily available in Australia, particularly for those living within large urban centres. However, this was not the case for breakfast due to the lack of both workplace canteens and affordable Chinese breakfast restaurants. Though no longer mandatory, most workplaces and schools in China provide a canteen on-site where employees can have meals for free or at a very low price, and the nutritional content of the meals provided by these canteens is often governed by local and provincial nutrition policies to ensure high quality food [32]. Previous research with Chinese migrants has also found that breakfast is the first meal to change post-migration and is much more likely to consist of Western-style foods; however, the reason for this change was previously poorly understood [9,33,34]. The participants in this study explained that they choose their breakfast based on convenience rather than taste preferences, as Chinese breakfasts take much longer to prepare. In China, their breakfast choices were also based on convenience, but the types of foods that were convenient in China are different to Australia. Contrary to previous research with Chinese immigrants, none of the participants reported skipping breakfast due to the unavailability of their preferred breakfast foods [35,36]. While breakfast was described as consisting of predominantly “Western” foods, all participants discussed cooking and consuming a Chinese-style evening meal. Another study exploring diet among Chinese American children found a similar pattern of primarily non-Chinese foods being consumed for breakfast, while immigrants maintained their dinner in the Chinese style [37]. Reasons for preferring to eat Chinese foods are numerous, but there have been some suggestions that maintaining connections with traditional eating patterns is a method of preserving cultural identity and connection with home [38]. On the other hand, dietary acculturation occurs due to the convenience of consuming Western style foods, which usually take less time to prepare than traditional Chinese dishes. This was not explicitly stated in the current study, but rather participants discussed familiarity with cooking methods and foods, as well as the fact that “Chinese people have Chinese stomachs”, as reasons for continuing to desire and seek out Chinese foods.

Due to diverse food environments in both China and Australia, the intersection of food cultures is bigger and, therefore, changes in diet post-migration are harder to notice, resulting in many of the participants feeling that their diets had changed very little after migration. Upon further exploration, it became clear that diets had in fact changed in substantial ways, but it was more as a result of recipe modification and the integration of Chinese and Western dietary behaviours and foods rather than a shift completely from Chinese to Western eating habits. This is also seen in multicultural societies, such as Singapore, where cross-cultural food practices are commonplace and members of the various ethnic and cultural groups routinely adapt recipes and incorporate ingredients from other food cultures in their cooking practices [39]. One of the ways in which diets changed most significantly for Chinese immigrants in this study, was in regard to who was responsible for food procurement and preparation. Most participants were not the primary food preparers when living in China, usually living with their parents or in-laws who would do most of the food shopping and cooking, so they may have lacked the necessary food literacy skills to take on this responsibility suddenly post-migration. Food literacy “includes knowledge, skills and behaviours required to plan, manage, select, prepare and eat food to meet needs and determine intake” [40] (pg 54). Higher levels of food literacy have been associated with healthier food consumption and self-control [41]. After migrating to Australia, most participants reported living in smaller insular families and having to take on the cooking and food shopping responsibilities, which contributed to consuming faster, easy-to-prepare meals for breakfast and lunch. However, the participants consistently expressed a preference for eating Chinese foods, and were willing to put in the extra effort to prepare evening meals in the Chinese style. In research with Chinese American families, similar preferences for continuing to consume Chinese food were observed and, in concordance with the current study, dinner was the meal where this cultural food consumption was preserved despite protests from children [34]. The study by Lv & Brown [34] found that adults’ eating habits are resistant to change because they established them when they were young, a sentiment that was echoed by some immigrants in our study, particularly those that were older in age. Food is very important in Chinese culture, both in the celebration of key events and for its use in preventing disease and maintaining health through the principles of traditional Chinese medicine [42] Other studies of dietary acculturation and food choice among Chinese migrants have found traditional Chinese medicine to be an important influence on dietary habits, particularly in maintaining Chinese traditional diets, but we did not find this to be an important theme in the current study [9,31,43]. Rather, personal health beliefs and knowledge gained through their own life experience and from family seemed more influential. While it is clear that consuming Chinese food continues to be important for immigrants, the way in which they maintain their cultural eating habits changes significantly post-migration due to changes in lifestyle and food environment.

Findings from our study indicate that after moving to Australia, Chinese people start to consume more snacks, which is less consistent with traditional Chinese eating patterns of three main meals with little between-meal snacking and much more consistent with Australian eating patterns [42,44]. Reasons why snacking behaviours increased after migration, included perception that food safety standards were higher in Australia compared to China, so participants were more willing to try pre-packaged Australian snacks, as well as the fact that snack foods did not have to be cooked, so it reduced the barriers that come with cooking unfamiliar food ingredients. China has faced a number of food safety issues over the years, which has led to consumer distrust in food products and food safety in general [45]. A systematic review also found food safety to be a determinant of food choice for Chinese people in recent times [43]. Regardless of the reason for increased snacking behaviour, it represents an area of concern for the health of Chinese immigrants, because snacking has been shown to increase risk of weight gain and cardiometabolic disease among both non-immigrant and immigrant populations [46,47,48]. Increased snacking combined with Westernized, convenient breakfast foods could contribute to the development of chronic diseases post-migration, and should be considered key targets for dietary interventions in Chinese immigrant populations.

### 4.1. Strengths and Limitations

There are some limitations to the present research, which include convenience sampling, which makes it difficult to determine if this group is representative of the Chinese immigrant population in Australia. Additionally, a large proportion of participants originated from coastal areas of China, with many consuming a lot of seafood prior to migrating, which might be different for people from other areas of China. As China has a diverse food geography and people from different regions consume vastly different traditional diets, migrants from other areas of China might have different experiences of dietary acculturation than the current participants [49]. However, migrants are known to be a hard-to-reach population in research, and most studies on Chinese immigrants have utilized convenience or snowball sampling [31,50].

Strengths of the current study include the in-depth interviews with Chinese migrants, providing qualitative insights into the experience of dietary acculturation for this hard-to-reach cultural group [50]. Additionally, interviews were conducted in tandem by two researchers with bilingual language skills, allowing interviews to be conducted in the participant’s preferred language, which participants expressed made them feel comfortable. The language skills of the researchers also allowed for better interpretation of the meaning and knowledge of the specifics of Chinese culture and food being discussed by participants.

### 4.2. Health Implications and Future Research

The findings of this study have a number of key implications for future research as well as interventions targeting dietary habits and the health of immigrant populations. The current research identified that Chinese immigrants are potentially at risk of reduced dietary variety due to changed social structures of food procurement and preparation, as well as unfamiliarity with new foods, which is associated with increased chronic disease risk. Efforts should be made by health educators to target food literacy skills in Chinese immigrants and also to encourage immigrants to try new foods that are healthy, require minimal preparation, and are safe to eat (rather than just snack foods). While not all aspects of dietary acculturation contribute negatively to health, a shift towards Western dietary behaviours has been shown to negatively impact the health of immigrants [7]. As the changes in diet seem to be occurring more subtly in the current population, such that they are not as acutely aware of some of the changes they have made, education around how to maintain a healthful diet in the new food environment is warranted, as well as education about how to substitute ingredients without sacrificing nutritional value. Availability of Chinese food ingredients did not seem to be an issue for most of the participants as they were easily able to find substitutes for particular spices and vegetables to fit their Chinese recipes; however, the consumption of pre-packaged snacks and convenient breakfast foods represent opportunities for further exploration and research to determine how these dietary changes impact on the risk of chronic diseases among Chinese migrants. One positive finding of this study is that Chinese immigrants seem to cook and eat at home more often after migration. Eating at home has been associated with lower consumption of fat, sugar, and energy, as well as lowered risk of chronic diseases in the long-term [51,52]. However, food literacy levels and time for food preparation may impact on the quality of meals being prepared at home by Chinese immigrants. Further research is needed to determine the quality of home-cooked meals post-migration for Chinese immigrants, as well as to assess food literacy skills in this particular population.

## 5. Conclusions

This study contributes to a better understanding of how dietary acculturation is experienced by Chinese immigrants to Australia. Though diets of Chinese immigrants do change post-migration, many of these changes are in relation to the social structures of food preparation, so many aspects of their traditional diet are also maintained, but in a new way that integrates their new lifestyle with their preference for eating Chinese foods. Some purposeful dietary change occurs in relation to breakfast and snacking behaviours while recipe modification of traditional Chinese foods results in unacknowledged dietary changes. Changes towards more convenient, Western-style foods might lead to potentially unhealthy dietary changes in Chinese immigrants, which may contribute to an increased risk of cardiometabolic disease over time. Cultural identity and familiarity remained important factors influencing food choice post-migration and, especially with older migrants, some resistance to change or a longing to maintain Chinese eating habits was evident. Better understanding of why and how diets change can help in the development of appropriate public health interventions to ensure that immigrants can make the necessary changes to their diet in a way that does not compromise health and reduces the risk of chronic diseases.

## Figures and Tables

**Table 1 nutrients-14-03573-t001:** Interview guide for semi-structured interviews designed to explore participant’s perspectives of dietary change post-migration.

Type of Question	Questions and Prompts	Relationship to Satia-Abouta et al. (2002) [10] Model of Dietary Acculturation
Questions designed to help participants feel comfortable and open up as well as to gather some basic demographic information about participants.	Tell me about where you were living before moving to Australia. How long have you been in Australia? Do you like Australia? Are you working here? What type of work do you do? Did you move to Australia with your family? (examines dynamics of family relationships) Do you go to church or community activity groups? Tell me more…	Socioeconomic and demographic factors
Questions designed to understand current dietary habits	Please describe your eating habits in Australia. *Prompts: How do you purchase/obtain your food? Who does the shopping and cooking? Who does most of the cooking/food preparation in your household? What types of food do you eat? Do you eat out/take out or eat at home? (never/occasionally/frequently)*	Food procurement and preparation post-migration
	How do you feel about Australia’s food and food habits/culture?	Cultural factors post-migration
	Are you able to buy all the types of food that you want? Why or why not? *Prompt: What do you think are your barriers/enablers (cost, accessibility, taste preference) in purchasing the foods you want?*	Changes in environment factors affecting food procurement
Questions designed to understand previous dietary habits in China	How would you describe your eating habits in China? *Prompts: How do you purchase/obtain your food? Who does the shopping and cooking? Who does most of the cooking/food preparation in your household? What types of food do you eat? Do you eat out/take out or eat at home? (never/occasionally/frequently)*	Food procurement and preparation pre-migration
	How do you feel about China’s food and food habits/culture?	Cultural factors pre-migration
Questions designed to understand how dietary habits have changed post-migration	How would you describe your current eating habits compared with your eating habits when you were living in China?	Changes in dietary intake
	What do you think influenced this change in your eating habits?	
Questions designed to understand factors influencing food choices	How would you describe yourself in trying new food?	Psychosocial factors, diet and disease related knowledge, and taste preferences
	What influences your food/eating habits? Do you perceive your diet to be healthy? Why or why not? (comparing two patterns of eating)	

**Table 2 nutrients-14-03573-t002:** Demographic characteristics of Chinese immigrants completing interviews (*n* = 11).

	Total
*n*	11
Age (years), mean (±SD)	43.9 (16.1)
Years residing in Australia, mean (±SD)	4.3 (2.2)
Sex, female, *n* (%)	8 (72.7)
Marital status, *n* (%)	
Single	4 (36.4)
Married/Partner	6 (54.5)
Divorced	1 (9.1)
Level of education, *n* (%)	
University degree or higher	9 (81.8)
Certificate or Diploma	1 (9.1)
High School or leaving certificate	1 (9.1)
Employment status, *n* (%)	
Part-time	4 (36.4)
Studying	2 (18.2)
Self employed	1 (9.1)
Unemployed	1 (9.1)
Fully retired	2 (18.2)
Home/Family Care	1 (9.1)

## Data Availability

The data presented in this study are available on request from the corresponding author. The data are not publicly available due to ethics reasons.

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
