# Peer review of "How and Why Diets Change Post-Migration: A Qualitative Exploration of Dietary Acculturation among Recent Chinese Immigrants in Australia"

_nutrients, 2022, doi:10.3390/nu14173573_

Round 1

Reviewer 1 Report

Interesting paper on how and why diets change after migration. Chinese immigrants living in Western countries are at greater risk of cardiometabolic diseases than those living in China. Dietary acculturation has been implicated as a potential contributor, but little is known about why diets change after migration.

The purpose of this qualitative research study was to explore how and why diets change after migration for Chinese immigrants living in Australia.

The methodology used in semi-structured interviews is adequate to the stated objective, as is the diversity of the individuals interviewed.

The thematic analysis carried out is valid and the information collected is relevant, supporting the discussion and conclusions presented.

Author Response

Thank you very much for taking the time to review our manuscript.

Reviewer 2 Report

Did you notice any differences in responses by age? 

How/why did you decide on 11 people?  Did you determine a point of 'saturation' in the responses?  That is not stated in the manuscript and should be added. 

Can you also include the median value for age along with the mean in line 166 with the range?

(With such a small sample, it is hard to determine if the ages of your participants are clustered.  Adding median will help the reader assess that besides the mean and sd.)

In Table 2, you can probably put the n for male and n for females in the first row and take it out of the table itself. 

P.2., lines 48-50:  Break into two sentences.  No comma is necessary between ...country and because.

P.3, lines 130-133:  It is not clear to me what you mean by interviewing in tandem.  Did you alternate questions during the same interview? Or just that you were both present at the same interview and then alternated interviews?  

P.6., lines 179-182:  That is a really long sentence.  Perhaps also break into two. 

Family support seems an important factor.  Did you ask about household size?  If not, maybe list that as a limitation too. 

I think it might be helpful to add the gender (sex) and age of your participants for the quotes, e.g., Participant 11, female, age 48.

This is a needed piece of scholarship and the results will be useful.  Perhaps in the future, you can augment your studies with a larger sample size
